# Assessing COVID-19 vaccine hesitancy and barriers to uptake in Sub-Saharan Africa

Philip Wollburg [1✉], Yannick Markhof [1,2], Shelton Kanyanda[3] & Alberto Zezza [1]

## Abstract

**Background** Despite improved availability of COVID-19 vaccines in Sub-Saharan Africa, vaccination campaigns in the region have struggled to pick up pace and trail the rest of the world. Yet, a successful vaccination campaign in Sub-Saharan Africa will be critical to containing COVID-19 globally.

**Methods** Here, we present new descriptive evidence on vaccine hesitancy, uptake, last-mile delivery barriers, and potential strategies to reach those who remain unvaccinated. Our data comes from national high frequency phone surveys in six countries in East and West Africa with a total population of 415 million people. Samples were drawn from nationally representative samples of households interviewed in recent in-person surveys. Our estimates are based on a survey module harmonized across countries and are re-weighted to mitigate potential sample selection biases.

**Results** We show that vaccine acceptance remains generally high among respondents in Sub-Saharan Africa (between 95.1% and 63.3%) even though hesitancy is non-negligible among those pending vaccination. Many who are willing to get vaccinated are deterred by a lack of easy access to vaccines at the local level. Furthermore, social ties and perceptions as well as intra-household power relations matter for vaccine take-up. Among the unvaccinated population, radio broadcasts have widespread reach and medical professionals are highly trusted.

**Conclusions** Our findings highlight that creating a positive social norm around COVID-19 vaccination, messaging that leverages trusted and accessible information sources and channels, and more easily accessible vaccination sites at the community level are promising policy options to boost vaccination campaigns in the region and end the pandemic everywhere.

## Plain language summary

COVID-19 vaccine coverage in Sub-Saharan Africa is behind the rest of the world. As the region is home to nearly 1.2 billion people (15% of the world population), achieving high levels of COVID-19 vaccination in Sub-Saharan Africa is important to containing the pandemic globally. We conduct national phone surveys in six countries in East and West Africa to learn how to best promote COVID-19 vaccine uptake in the region. Our surveys focus on peoples' willingness to get vaccinated, barriers that prevent them from accessing COVID-19 vaccines, and strategies to reach out to those who have not been vaccinated yet. We find that vaccine acceptance is high but that poor access to vaccines at a local level prevents many from getting vaccinated. Our findings can help policymakers design more effective vaccination campaigns.

[1] Development Data Group, World Bank, Washington, DC, USA. [2] UNU-MERIT, United Nations University, Maastricht, Netherlands. [3] National Statistical Office, Zomba, Malawi. ✉email: pwollburg@worldbank.org

Starting in late 2020, the world has seen the largest vaccination effort in history[1]. In April 2023, over two years after the availability of the first COVID-19 vaccines, nearly 65 percent of the world population has been fully vaccinated for COVID-19. However, large regional disparities in COVID-19 vaccine coverage remain[2]. Sub-Saharan Africa (SSA), in particular, is trailing the rest of the world. As of April 2023, less than 30 percent of the population had received at least two doses. All but four countries in Sub-Saharan Africa (the small island states of the Seychelles and Mauritius as well as Rwanda and Liberia) remain adrift of the World Health Organization's (WHO) goal to fully vaccinate over 70 percent of the African population by June 2022[3].

This discrepancy is troubling. Leaving the poorest region in the world largely unprotected exacerbates existing health and economic inequities. Low COVID-19 vaccine coverage in Sub-Saharan Africa is also concerning from a global epidemiological perspective. Sub-Saharan Africa is home to almost 1.17 billion people, 15 percent of the world population. Low COVID-19 vaccination rates in this region compound the risk of continued mutation of the virus[4,5]. Increasing vaccine coverage in Sub-Saharan Africa while the current vaccines are still effective is therefore key to ending the pandemic everywhere[5,6].

A variety of reasons have been cited for why vaccination efforts in Sub-Saharan Africa are trailing their targets[7–16]. Shortages in vaccine supplies, the accessibility of vaccination sites, and vaccine hesitancy are the main hypotheses discussed to explain the low coverage[7,17]. However, there is a conspicuous lack of systematic, cross-country comparable, and up-to-date evidence on these issues. In this study, we use data from recent national phone surveys in six countries in East and West Africa – Burkina Faso, Kenya, Malawi, Nigeria, Tanzania, and Uganda – to contribute new insights to a growing body of literature on vaccine hesitancy, uptake, barriers of access at the community level, and possible promoters of vaccine demand in the region.

Our contributions are fourfold. First, unlike many existing studies that examine specific population groups or regions within countries, our data is national in scope and was collected in collaboration with the respective national statistical agency of each country we study (Burkina Faso, Kenya, Malawi, Nigeria, Tanzania, and Uganda) with samples drawn from large, nationally representative sampling frames. Second, data collection efforts were harmonized to a high degree between countries by incorporating a newly designed set of vaccination-related questions in the national phone surveys. Our insights are therefore cross-country comparable and cover six countries with a population of approximately 415 million people, 35% of the population of Sub-Saharan Africa. Third, the evidence we present is timely, collected between November 2021 and August 2022. It adds to and updates existing evidence on issues such as vaccine hesitancy from before the start of vaccination for the general population in Sub-Saharan Africa[18–22]. Fourth, our analysis simultaneously addresses a broad range of issues and information gaps that matter for vaccination efforts. It provides microlevel evidence on all five dimensions of vaccine hesitancy, *confidence, complacency, convenience, communication*, and *context*, last-mile delivery barriers, and potential strategies to reach those who remain unvaccinated[23]. These insights come at a crucial moment for vaccination campaigns which have struggled to pick up pace.

We find that in our study countries a majority remains willing to get vaccinated. The main barriers to vaccine access are country-specific but commonly relate to the ease with which vaccines can be accessed within communities. Therefore, it is indispensable that vaccination sites become more widespread at the local level. In addition, we find that communication campaigns leveraging trusted vaccine ambassadors and emphasizing the health benefits of COVID-19 vaccines are promising strategies to encourage vaccine uptake.

## Methods
**Data**. We use data from High Frequency Phone Surveys (HFPS) in six countries in East and West Africa – Burkina Faso, Kenya, Malawi, Nigeria, Tanzania, and Uganda. The surveys were conducted by countries' national statistical agencies in collaboration with the World Bank. Since 2020, the HFPS have collected cross-country comparable longitudinal data on a wide range of topics, including the impacts of COVID-19 on households and individuals. They also provide extensive demographic information through the administration of a roster of all household members in each survey round.

Some rounds of the HFPS include a module on COVID-19 vaccination. The cross-sectional data we use in this study was collected between November 2021 and August 2022: Burkina Faso (April–May 2022), Kenya (November 2021–March 2022), Malawi (February 2022), Nigeria (December 2021–January 2022), Tanzania (December 2021), and Uganda (August 2022).

**Sampling and sample representativeness**. The HFPS are re-contact surveys with national scope whose samples are drawn from nationally representative samples of households interviewed in recent in-person surveys.

The face-to-face surveys from which the phone survey samples were drawn were sampled using stratified two-stage clustered sampling, with the first sampling unit the enumeration area, the second the household, and stratification by urban/rural and sub-national administrative units. Enumeration areas were drawn with probability proportional to size (PPS) and the households were drawn with simple random sampling (SRS). The face-to-face surveys are nationally and sub-nationally representative. The face-to-face surveys used as sampling frames for the phone surveys are the Burkina Faso Enquête harmonisée sur les conditions de vie des ménages 2018/19, the Kenya Integrated Household Budget Survey 2015/16, the Malawi Integrated Household Panel Survey 2019, the Nigeria General Household Survey – Panel 2018/19, and the Uganda National Panel Survey 2019/20[24]. The face-to-face surveys collected contact information of the households interviewed during fieldwork. Households with access to a phone provided their phone number(s) while for those without a phone the survey implementers attempted to record a reference contact's phone number (such as a neighbor, friend or family member).

The phone survey samples were drawn from all available phone numbers as collected in the face-to-face surveys (Table 1).

In Kenya, an additional sample of households was drawn via random digit dialing (RDD). This consisted in creating a list of 92,999,970 randomly ordered phone numbers using a random number generator from the 2020 Numbering Frame of the Kenya Communications Authority. The phone numbers were from three networks: Safaricom, Airtel and Telkom (as a result, there are more phone numbers than Kenya's population size). In the next step, introductory text messages were sent to 5000 randomly selected numbers and 4075 were determined to be operational and formed the final sampling frame[25]. Coverage rates (share of face-to-face households for whom a contact phone number was available) varied between countries, ranging from 41% in Kenya to 99% in Nigeria (98% in Burkina Faso, 77% in Uganda, 73% in Malawi; information not available for Tanzania and not applicable Kenya RDD sample). Coverage rates in Nigeria, Burkina Faso, Uganda, and Malawi far exceed phone penetration rates because of the inclusion of reference contacts outside of the household[26,27].

**Table 1 Sample size, coverage and response rates.**

| Sample of households: | Burkina Faso | Kenya (KIHBS) | Kenya (RDD) | Malawi | Nigeria | Tanzania | Uganda |
|---|---|---|---|---|---|---|---|
| A. Pre-COVID F2F Survey / RDD frame | 7010 | 21,773 | 92,999,970 | 3181 | 4976 | 12,812 | 3098 |
| B. Phone numbers available | 6877 | 9007 | 92,999,970 | 2337 | 4934 | | 2386 |
| C. Attempted to contact | 2199 | 9007 | 5,000 | 2337 | 4440 | 5750 | 2386 |
| D. Non-contact | 248 | - | 925 | 608 | 1288 | 3499 | 98 |
| E. Successful contact | 1951 | | | 4075 | 1729 | 3152 | 2251 | 2288 |
| F. Incomplete interviews | 104 | - | 3000 | 282 | 230 | 58 | 406 |
| G. Complete interviews | 1847 | 4561 | 1075 | 1447 | 2922 | 2193 | 1882 |
| Coverage rate (B/A) | 98% | 41% | N/A | 73% | 99% | - | 77% |
| Response rate (G/[G + D + F]) | 84% | 51% | 22% | 62% | 66% | 38% | 79% |

Sampling frame in Kenya consisted of re-contact sample from the Kenya Integrated Household Budget Survey 2015/16 (KIHBS) and a sample obtained from random digit dialing (RDD). The Kenya RDD frame includes numbers from three operators. Coverage rate for Kenya (RDD) not applicable. Response rate for Kenya (KIHBS) is based on Attempted to contact / Complete Interviews.

Response rates, calculated as Complete interviews / [Complete interviews + Non-contact + Incomplete Interviews] (which is equivalent to Complete interviews / Attempted to contact) are 22% in Kenya RDD, 38% in Tanzania, 51% in Kenya KIHBS, 62% in Malawi, 66% in Nigeria, 79% in Uganda, and 84% in Burkina Faso (Table 1).

Incomplete coverage (i.e., coverage rate less than 100%) means that there is some portion of the population that cannot be included in the phone surveys. Similarly, non-response (i.e. response rate less than 100%) means that not all contacted households ended up participating in the survey. Given incomplete coverage and non-response, the raw phone survey samples may not be fully representative of the general population. This is because non-covered and non-responding households may be different from interviewed households, such that the sample of interviewed households would not be representative of all households. To mitigate these concerns, sampling weights were recalibrated using propensity score and post-stratification methods. With propensity score reweighting, information on interviewed and non-interviewed households is used to give more weight to households that are more representative of the general population of households. With post-stratification, weights are scaled to sum up to known population totals at a suitable sub-national level (e.g. admin1 regions)[28]. These reweighting exercises have been shown to considerably mitigate the effects of sample selection from under-coverage and non-response in a study that uses several of the surveys included in this study (Malawi, Nigeria, Uganda)[29]. For the RDD sample, there is no baseline information on covered and non-covered households so that the extent of recalibration of the RDD survey weights is more limited and these estimates might be less representative than the other estimates.

In each household, one main respondent over the age of 15 was interviewed, who was selected to be knowledgeable of the affairs of the household and its members to provide reliable responses. This purposive selection overrepresents certain population groups – household heads, men, better educated, older individuals. We show the profiles of phone survey respondents in our sample and of the general adult population (based on pre-COVID-19 face-to-face household surveys) in Supplementary Table 16. Weight recalibration has been shown to fail to fully overcome the effects of this purposive respondent selection[30]. In all, while we use recalibrated sampling weights to improve the representativeness of our estimates, our estimates cannot be considered fully nationally representative. Estimates from study countries with high coverage and response rates are likely closer to nationally representative than those with lower coverage and response rates and RDD-based estimates.

The target sample size for the HFPS was set to be sufficient to detect a 10 percentage point change in the key indicators (COVID-19 knowledge and behavior and labor market impacts) in between rounds with 90% power and 95% confidence at the national level. Detecting changes across survey rounds requires much larger samples than accurate estimation within a single round as in the case of our study. Table 1 summarizes the resulting sample sizes for each country included in our study.

**Survey instrument and variables**. A harmonized survey module on COVID-19 vaccination was implemented in all six study countries for the purpose of this study. Participants were already familiar with the HFPS and were informed that the goal of the vaccine module was to understand people's attitudes towards COVID-19 vaccines and that this information would not be used to determine their eligibility to receive a COVID-19 vaccine or to provide them with access to a COVID-19 vaccine.

To study vaccine hesitancy and uptake in a systematic and comprehensive manner, we distinguish three groups in the population: (i) those already vaccinated, (ii) those who are willing to get vaccinated but are yet unvaccinated, and (iii) those who are hesitant to get vaccinated. We define hesitancy in our data as those respondents who are either unwilling to get vaccinated or uncertain about their decision. Analogously, we categorize as willing those who have either already been vaccinated or are willing to do so.

Our analysis draws on the questions summarized in Supplementary Note 1. These questions group into six broad themes. First, we ask about vaccine uptake among respondents and ask those who have already been vaccinated about the vaccination process. The second theme relates to vaccine acceptance among those who have not been vaccinated yet. These questions gauge the extent of vaccine hesitancy among respondents and the main reasons for their vaccine attitudes. The third theme covers the barriers that keep respondents from getting vaccinated with a particular focus on those who have not been vaccinated yet despite being willing to. Fourth, we ask respondents about their main sources and channels of information on COVID-19 vaccines to identify possible communication strategies for national vaccination campaigns. Theme five is devoted to potential ambassadors that could promote vaccination among respondents. Finally, the sixth theme turns to the social context in which vaccine attitudes are formed and looks at the transmission of vaccine attitudes and uptake decisions within communities and households.

**Survey implementation**. The enumerators recruited for our survey were selected out of a pool of existing enumerators with experience conducting (LSMS-ISA) household surveys. They had therefore undergone previous (LSMS) training and were intimately familiar with LSMS-style surveys. In some of the

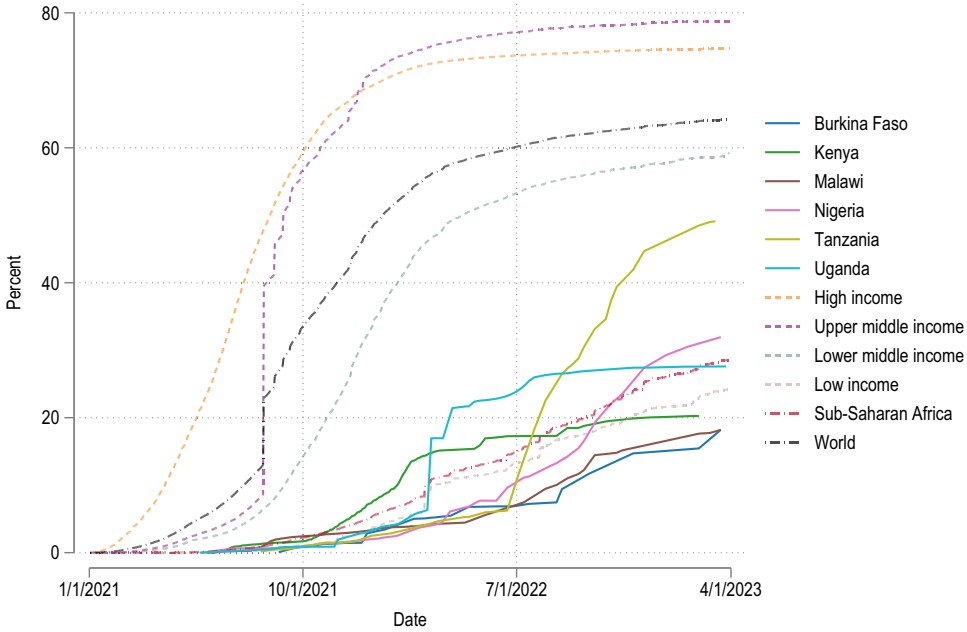

**Fig. 1 Share of the population fully vaccinated over time.** Source: Mathieu et al.[2]. Blue solid line = Burkina Faso, green solid line = Kenya, brown solid line = Malawi, pink solid line = Nigeria, chartreuse solid line = Tanzania, cyan solid line = Uganda, orange dashed line = high income, purple dashed line = upper middle income, blue-grey dashed line = lower middle income, rose dashed line = low income, red dash-dotted line = Sub-Saharan Africa, black dash-dotted line = World.

countries, enumerators furthermore had previous experience conducting surveys over the phone. All interviewers received three days of standardized training ahead of the first round of the HFPS and a one-day follow-up training in preparation of each new survey round.

To minimize human error and ensure the high quality of data collected, answers were recorded using computer-assisted telephone interviewing (CATI) and regular audio-audits were performed. Further, the implementation of each survey was preceded by three days of piloting the questionnaire, CATI technology, survey protocols, and monitoring mechanisms in a sample matching our target population.

**Statistics and reproducibility.** Reported estimates are population-weighted means with 95% confidence intervals, with recalibrated phone survey weights used throughout. Figure 1 uses data from the Our World in Data COVID-19 vaccination dataset and presents country and region totals[2,31]. The full results are presented in Supplementary Tables 1–15.

**Ethical compliance.** Each phone survey was implemented by the respective national statistical office (NSO). The NSO conducts the survey as the sole official statistical authority in the country and in accordance with the respective National Statistical Act, which exempts the NSO from institutional ethics approvals. Informed consent was received from all survey respondents in each country and any identifying information anonymized. The World Bank does not require institutional ethics approval for household surveys that are partly or fully financed by the World Bank, including the national phone surveys in Burkina Faso, Kenya, Malawi, Nigeria, Tanzania, and Uganda that inform our research.

**Reporting summary.** Further information on research design is available in the Nature Portfolio Reporting Summary linked to this article.

## Results

**A majority is willing to get vaccinated, but COVID-19 vaccine hesitancy should not be dismissed.** Overall, we find that estimated acceptance rates for COVID-19 vaccines in Sub-Saharan Africa remain at high levels of acceptance, similar to those reported in late 2020. These high levels of acceptance contrast with low uptake of COVID-19 vaccines in Sub-Saharan Africa compared to the rest of the world (Fig. 1).

In Kenya vaccine acceptance is near universal (95.1%, 95% confidence interval (CI) 93.4–96.9%), likely reflecting the issuance of vaccination requirements for public services and places by the Kenyan government in December 2021. Similarly, in Uganda, we estimate high vaccine acceptance at 90.8% (CI 88.9–92.8%) as of August 2022. In Nigeria, almost four in five people (78.4%, CI 75.8–80.9%) would accept or have already accepted to be vaccinated for COVID-19. Vaccine acceptance is also above the WHO's envisioned 70% threshold in Malawi (75.1%, CI 71.1–79.1%) and Burkina Faso (74.4%, CI 71.5–77.2%). At the same time, vaccine hesitancy is notably higher in Tanzania where less than two-thirds of the population (63.3%, CI 60.2–66.3%) are willing to get vaccinated. This is consistent with the COVID-19-sceptic stance the Tanzanian government initially took during the pandemic[32]. Our point estimates suggest higher hesitancy among women than men (significant in Kenya, Nigeria, and Tanzania) and in urban areas compared to rural areas (significant in Burkina Faso, Kenya, Nigeria, and Uganda, Supplementary Table 1).

Among those who are still unvaccinated, the main group for vaccination campaigns to reach out to, vaccine hesitancy is expectedly higher (Fig. 2). In Kenya and Nigeria, reported vaccine acceptance still stands at 86.4% (CI 81.8–91.0%) and 70.5% (CI 67.3–73.8%) among the unvaccinated. However, this figure is lower in Malawi (59.1%, CI 53.5–64.7%), Burkina Faso (57.3%, CI 53.0–61.5%), and Tanzania (56.5%, CI 53.1–59.8%). In Uganda, where a larger share of survey respondents had already been vaccinated, fewer of the remaining respondents were still willing to be vaccinated (34.4%, CI: 25.9 to 42.9). In all countries,

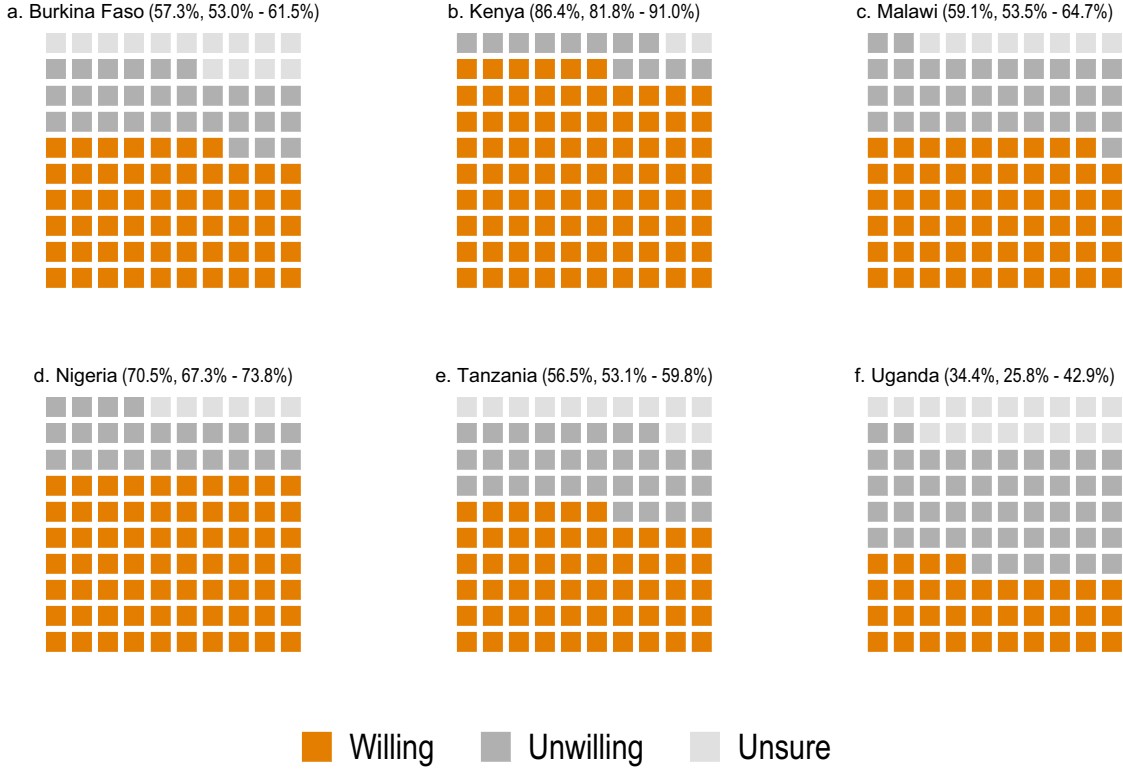

**Fig. 2 Vaccine acceptance among the unvaccinated.** Acceptance of COVID-19 vaccines among the population that is currently unvaccinated. Point estimates of the average acceptance rate and 95% confidence intervals in parentheses above each panel. Burkina Faso (**a**, $n = 1157$ respondents), Kenya (**b**, $n = 2018$ respondents), Malawi (**c**, $n = 863$ respondents), Nigeria (**d**, $n = 2010$ respondents), Tanzania (**e**, $n = 1791$ respondents), Uganda (**f**, $n = 252$ respondents). Orange squares = willing, dark grey squares = unwilling, light grey squares = unsure.

the majority of those we classify as hesitant report being unwilling to be vaccinated rather than uncertain of their decision.

**Ease of access to vaccines remains a key barrier.** Our data confirms that self-reported knowledge about the start of vaccination campaigns is very high and does not seem to prevent vaccine uptake. Among those currently unvaccinated, knowledge about the start of vaccination campaigns in their country is almost universal in Kenya (99.9%, CI 99.9–100%), Malawi (99.3%, CI 98.5–100%) and Tanzania (96.3%, CI 95.1–97.5%), very high in Burkina Faso (92.5%, CI 90.4–94.5%), and slightly lower in Nigeria (82.2%, CI 79.9–84.6%).

What is preventing those who are willing to be vaccinated from doing so? Reasons vary between countries but typically relate to the ease of access to vaccinations (Fig. 3 and Supplementary Table 2). In Nigeria, almost four in ten unvaccinated people do not know how to get vaccinated (39.6%, CI 34.8–44.5%). In Malawi, Nigeria, and Tanzania, a substantial share reports prohibitively long distances to the nearest vaccination point (Malawi: 13.6%, CI 8.8–18.4%; Tanzania: 12.2%, CI 9.2–15.1%; Nigeria: 12.0%, CI 9.2–14.7%) or a lack of available vaccines (Malawi: 18.6%, CI 12.5–24.7%); Nigeria: 9.6%, CI 7.0–12.3%; Tanzania: 9.1%, CI 6.7–11.6%). In Tanzania and Nigeria, traveling to the vaccination site also commonly clashes with work commitments (Tanzania 28.0%, CI 24.1–32.0%; Nigeria 11.7%, CI 8.4–15.0%). In contrast, fear of potential side effects of the vaccine is only a widespread concern in Malawi (35.2%, CI 28.0–42.4%). Structural issues such as the availability of vaccines and distance to the nearest vaccination point are more frequently reported in rural areas whereas in urban areas, work commitments commonly stand in the way of getting vaccinated. Furthermore,

there is a gender divide, with women commonly citing domestic commitments and medical reasons while men frequently mention work commitments (Supplementary Tables 3, 4).

In Kenya, a similar question on the anticipated barriers of access was asked (Supplementary Table 5). While the largest group cites no barriers of access (37.9%, CI 31.4–44.5%), substantial shares are deterred by crowded vaccination sites (27.8%, CI 21.6–33.9%), a lack of vaccines in sufficient numbers (25.8%, CI 20.0–31.6%), and long distances to the nearest vaccination point (23.5%, CI 17.7–29.2%).

Insufficient ease of access to COVID-19 vaccines is also reflected in the vaccination points used (Supplementary Table 6). The majority of respondents was vaccinated at a medical site such as a health center (Burkina Faso: 45.4%, CI 39.2–51.7%; Tanzania: 45.1%, CI 38.2–52.1%; Uganda: 41.7%, CI 38.2 to 45.2; Nigeria: 30.7%, CI 25.6–35.8%; Malawi: 23.4%, CI 17.6–29.2%; Kenya: 18.3%, CI 14.5–22.1%) or a hospital (Kenya: 59.9%, CI 55.2–64.6%; Malawi: 43.6%, CI 37.1–50.1%; Uganda: 34.0; CI 30.6 to 37.4%; Tanzania: 31.8%, CI 25.4–38.2%; Nigeria: 29.5%, CI 24.8–34.2%; Burkina Faso: 13.7%, CI 9.3–18.2%). Somewhat fewer people were vaccinated at mass vaccination sites (Burkina Faso: 35.1%, CI 29.2–40.9%; Malawi: 20.0%, CI 13.8–26.3%; Uganda: 16.5%, CI 13.9–19.0%; Nigeria: 12.6%, CI 9.3–15.9%; Tanzania: 11.9%, CI 7.5–16.3%; Kenya: 8.9%, CI 6.7–11.1%) and smaller, local vaccination sites such as pharmacies, clinics, religious centers, senior homes, or people's work location play only a minor role.

**Beliefs about the health benefits and risks of vaccines drive vaccine acceptance and uptake.** A large share of the vaccinated population reports that protecting their own health was the primary and only reason to get vaccinated (Malawi: 89.9%, CI

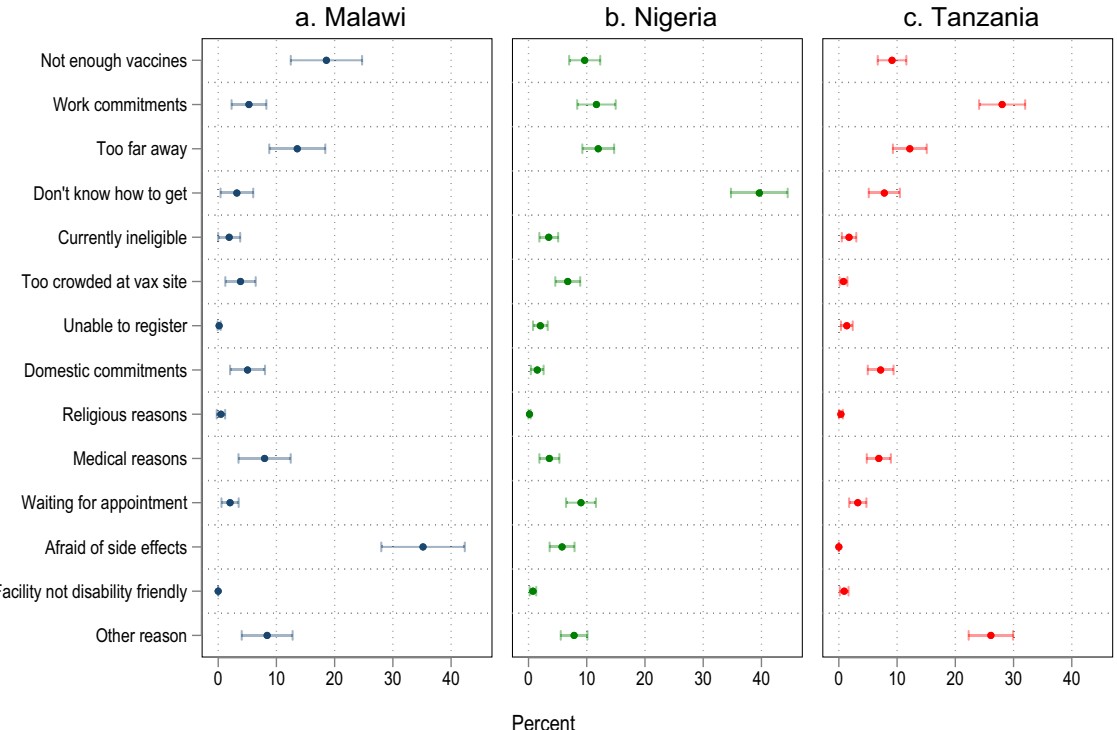

**Fig. 3 Reasons for pending vaccination despite being willing to get vaccinated.** The dots represent estimated means, the bars around them the 95% confidence interval of the estimate. Malawi (**a**, *n* = 493 respondents), Nigeria (**b**, *n* = 1098 respondents), Tanzania (**c**, *n* = 992 respondents).

86.5–93.3%; Tanzania: 76.8%, CI 70.8–82.9%; Uganda: 65.1%, CI 61.7 to 68.6%; Nigeria: 64.1%, CI 59.5–68.7%; Burkina Faso: 60.8%, CI 55.1–66.5%; Kenya: 22.7%, CI 18.4–26.9%; Supplementary Table 7). This makes a strong case for vaccination campaigns to emphasize the health benefits of vaccination when aiming to increase COVID-19 vaccine take-up. Other motivations commonly cited for getting vaccinated are to protect the health of others in Kenya (68.6%, CI 64.1–73.1%) and Burkina Faso (26.7, CI 21.4–31.9%) and government mandates in Nigeria and Uganda (Nigeria: 18.7%, CI 14.8–22.5%; Uganda: 23.3%, CI 20.3–26.4%). In Kenya, substantial numbers also got vaccinated because they considered it "the right thing to do" (29.7%, CI 25.3–33.1%).

Concerns about the vaccine's side effects are the main reason for vaccine hesitancy in all countries studied (Kenya: 86.0%, CI 77.8–94.1%; Malawi: 45.8%, CI 38.3–53.2%; Tanzania: 41.3%, CI 36.4–46.2%; Burkina Faso: 36.5%, CI 29.9–43.2%; Uganda: 29.6%, CI 19.0–40.2; Nigeria: 20.8%, CI 15.7–26.0%; Supplementary Table 8). In Kenya, worries about the safety of vaccines are also common (45.2%, CI 27.2–63.2%). Other reasons for hesitancy reflect a lack of confidence, either in vaccines in general or the COVID-19 vaccines in particular: Among vaccine hesitant individuals, one in three in Burkina Faso (33.2%, CI 25.8–40.6%), 15.1% in Malawi (CI 9.6–20.6%), 10.5% in Tanzania (CI 7.5–13.5%), 9.5% in Kenya (CI 3.3–15.8%), 4.7% in Nigeria (CI 2.6–6.8%), and 1.3% in Uganda (CI 0.0–2.9%) say they do not think the vaccine works. In Tanzania, Malawi, and Uganda, around one in five hesitant people say that they generally do not trust vaccines. Finally, a non-negligible share of the hesitant also simply do not regard getting vaccinated as a high enough priority (Malawi: 21.6%, CI 15.0–28.3%; Uganda: 13.4%, CI 6.5–20.2%; Tanzania: 10.9%, CI 7.9–13.9%; Nigeria: 11.0%, CI 6.8–15.3%); Burkina Faso: 8.7%, CI 5.3–12.1%).

**Medical professionals are widely trusted as vaccine ambassadors.** The most common information sources on COVID-19 vaccination are medical professionals (doctors, nurses, pharmacists, and other health workers) and the media. Medical professionals are the most trusted information source in Burkina Faso (45.2%, CI 41.0–49.4%), Nigeria (28.7%, CI 25.7–31.7%) and Uganda (30.4%, CI 27.3–33.4%) and also a highly trusted information source in Malawi (34.3%, CI 30.1–38.6%) (Supplementary Table 9). Notably, those willing to be vaccinated are more likely to trust medical professionals when it comes to COVID-19 information than the hesitant, except in Uganda. Conversely, the hesitant are relatively more likely to cite the media over medical professionals as their most trusted source of information on COVID-19 vaccines (Fig. 4 and Supplementary Table 10). This may relate to an association between vaccine hesitancy and misinformation spread through media channels that previous studies found[8,16,33]. In Uganda, around a quarter of respondents consider local government authorities as their most trusted source of information.

Leveraging trusted figures as ambassadors of COVID-19 vaccination could be a viable strategy to promote vaccine uptake. Across the countries we study, a large but varying share of the hesitant report that they would be more likely to get vaccinated if it was recommended to them by a "vaccine ambassador" (Tanzania: 77.3%, CI 73.1–81.4%; Uganda: 64.4%, CI 51.8 to 77.0%; Nigeria: 58.2%, CI 52.3–64.1%; Kenya: 44.0%, CI 25.2–62.9%; Burkina Faso: 42.1%, CI 35.8–48.3%; Malawi: 36.1%, CI 27.2–45.0%). When asked about what vaccine ambassadors could encourage them to get vaccinated, the hesitant most frequently mention medical professionals (Uganda: 52.7%, CI 40.0 to 65.4%; Nigeria: 40.1%, CI 34.5–45.7%; Kenya: 34.8%, CI 15.3–54.3%; Burkina Faso: 34.5%, CI 28.1–40.9%; Tanzania: 31.9%, CI 27.2–36.5%; Malawi: 26.6%, CI 19.1–34.2%) and family and other community members (Nigeria: 42.7%, CI 36.9–48.6%;

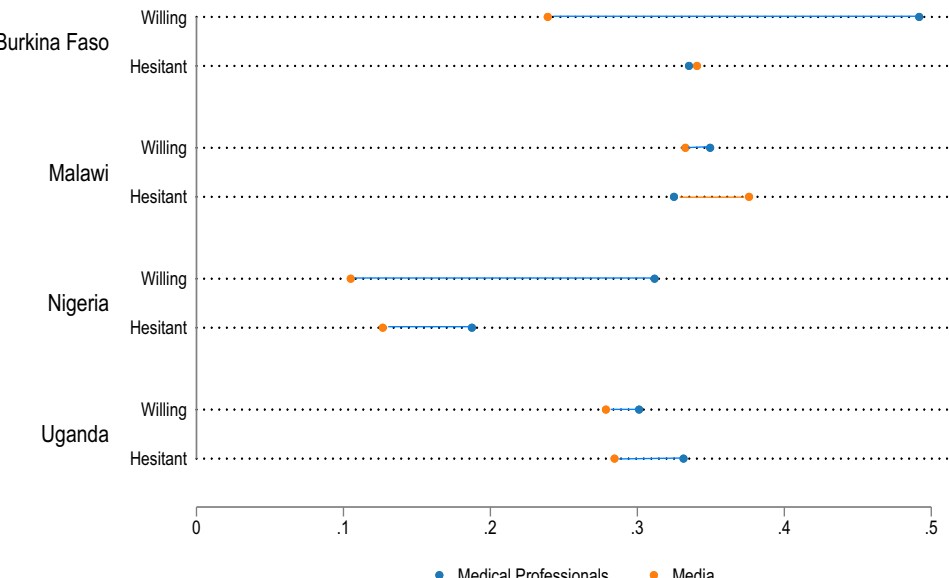

**Fig. 4 Most trusted information source on COVID-19 vaccines by vaccine acceptance.** Plotted are the mean estimates for the two most common options, media and health professionals. The lines connecting the dots for each country and vaccine acceptance status represent the gap between how commonly the media is cited as the most trusted information source and how often health professionals are listed instead. A blue line indicates that health professionals are more commonly cited, an orange line that the media is more often mentioned. Burkina Faso (Willing: n = 1261 respondents; Hesitant: n = 483 respondents), Malawi (Willing: n = 1077 respondents; Hesitant: n = 366 respondents), Nigeria (Willing: n = 2021 respondents; Hesitant: n = 477 respondents). Uganda (Willing: n = 1716 respondents; Hesitant: n = 156 respondents). Blue dots = share naming medical professionals, orange dots = share naming media, blue solid line = medical professionals more often trusted than media, orange solid line = media more often trusted than medical professionals.

Malawi: 29.9%, CI 21.2–38.6%; Burkina Faso: 28.3%, CI 21.5–35.1%; Kenya: 17.6%, CI 8.2–27.1%; Tanzania: 12.1%, CI 8.9–15.4%; Supplementary Table 11).

**Media outlets such as radio and television reach large shares of the population with information on COVID-19 vaccines.** Across countries, we estimate that more than half of the population rely on radios to receive their most trusted information on COVID-19 vaccines (Uganda: 70.6%, CI 67.7–73.5%; Burkina Faso: 67.0%, CI 63.1–70.9%; Nigeria: 58.8%, CI 55.7–61.9%; Malawi: 51.1%, CI 47.0–55.3%; Supplementary Table 12). This emphasizes the role of radio broadcasting as an effective medium of information transmission that has wide dissemination across Sub-Saharan Africa and among different population groups[34]. Other important channels of vaccine information transmission are in-person interactions (Malawi: 61.7%, CI 57.3–66.0%; Uganda: 52.7%, CI 49.4 to 56.1%; Burkina Faso: 44.9%, CI 40.9–48.9%; Nigeria: 42.8%, CI 39.9–45.7%) and, to a lesser extent, television (Burkina Faso: 36.1%, CI 32.3–39.9%; Uganda: 25.6%, CI 22.6 to 28.5; Nigeria: 25.2%, CI 22.2–28.3%; Malawi: 4.5%, CI 2.8–6.2%).

**The social context informs vaccine attitudes.** Beliefs about the attitudes and norms prevalent in one's social circle have been shown to influence individual attitudes and behavior[35,36]. Notably, people are found to often misperceive the true attitudes of those around them and correcting these beliefs can change behavior[35]. We explore perceived vaccine acceptance in one's community by asking respondents to estimate how many members of their community out of ten would be willing to be vaccinated.

Across our study countries, respondents estimate vaccine acceptance levels to be lower than the national acceptance levels in our data (Fig. 5 and Supplementary Table 13). Furthermore, the margin by which vaccine acceptance is underestimated is large (Burkina Faso: 43.2% vs. 74.4%, two-sided t-test $F$ (531) = 453.78, $p < 0.001$; Malawi: 47.7% vs. 75.1%, $F$ (247) = 216.38, $p < 0.001$; Nigeria: 59.1% vs. 78.4%, $F$ (516) = 203.03, $p < 0.001$; Tanzania: 43.6% vs. 63.3%, $F$ (899) = 110.07, $p < 0.001$; Uganda: 61.1% vs. 90.8%, $F$(891) = 634.41, $p < 0.001$). While both the hesitant and willing on average perceive vaccine acceptance in their community to be below the national average, this difference is much more pronounced among the hesitant. Discrepancies between the hesitant and willing are largest in Nigeria (39.1 percentage points), substantial in Burkina Faso (26.6 pp), Tanzania (25.9 pp), and Malawi (20 pp) and somewhat smaller in Uganda (14.2 pp). This suggests a correlation between social perceptions and personal vaccine attitudes.

While community dynamics seem to play a role in the formation and transmission of COVID-19 vaccine attitudes, social processes and dynamics within households matter as well (Fig. 6 and Supplementary Table 14). According to the respondents, the household head makes the vaccination decision on behalf of the household members in 69.0% of households in Nigeria (CI 65.5–72.6%), 44.4% in Tanzania (CI 41.1–47.7%), 42.3% in Burkina Faso (CI 37.9–46.8%), 39.4% in Malawi (CI 35.4–43.4%), and 36.8% in Uganda (CI 33.7–40.0%). In contrast, vaccination decisions are left to individual household members among 50.9% of households in Burkina Faso (CI 46.7–55.2%), 42.0% in Malawi (CI 37.6–46.5%), 42.0% in Uganda (CI 38.8–45.2%) 37.1% in Tanzania (CI 34.0–40.3%), and 18.3% in Nigeria (CI 15.2–21.5%). Power dynamics within the household can thus mean that vaccination is not an individual decision.

Given the role of social interactions in determining vaccine attitudes and convincing the hesitant, we lastly consider the willingness among those already vaccinated to act as potential ambassadors of COVID-19 vaccination (Supplementary Table 15).

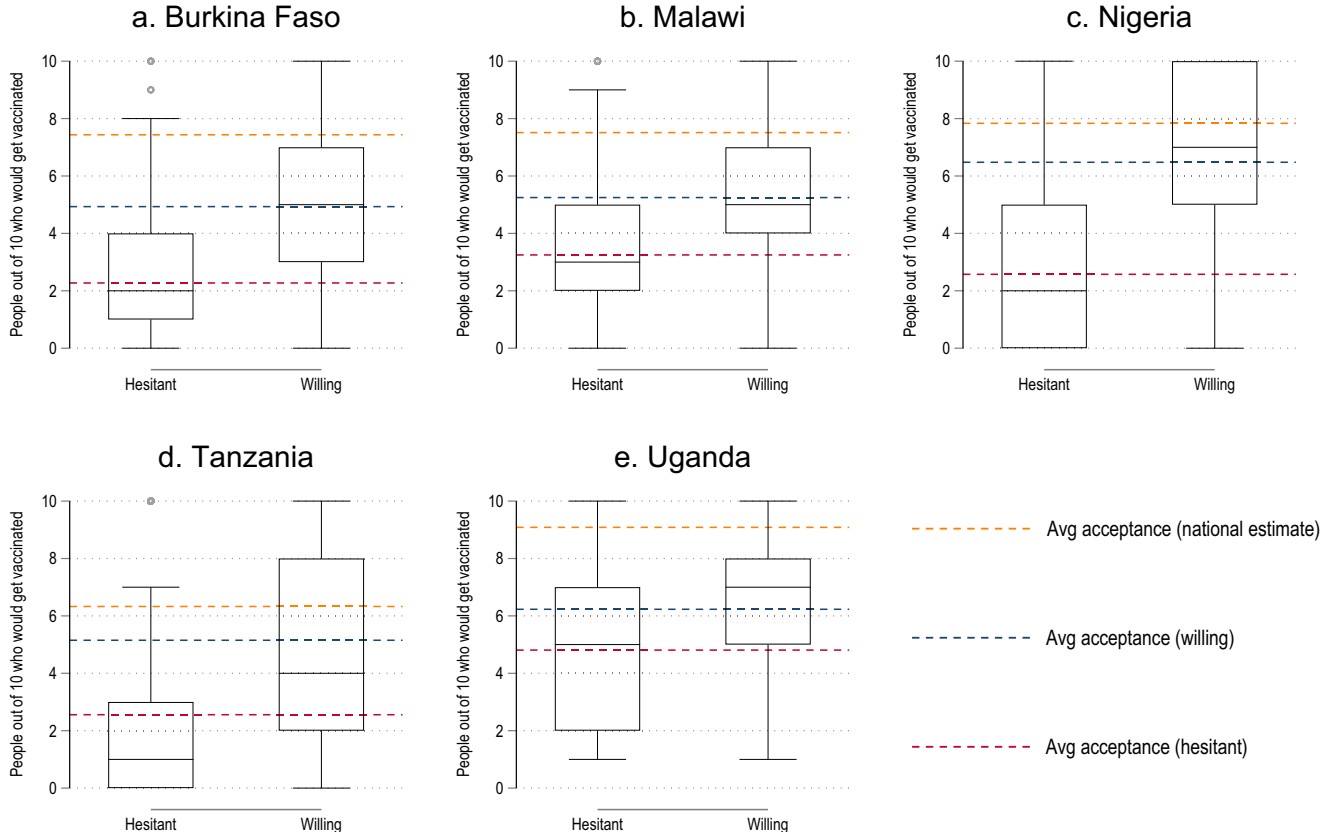

**Fig. 5 Perceived vaccine acceptance within their community among the hesitant and willing.** Number of people out of 10 that respondents think would get vaccinated for COVID-19 in their community. Burkina Faso (**a**, National estimate: $n = 1847$ respondents; Willing: $n = 1162$ respondents; Hesitant: $n = 389$ respondents), Malawi (**b**, National estimate: $n = 1447$ respondents; Willing: $n = 972$ respondents; Hesitant: $n = 315$ respondents), Nigeria (**c**, National estimate: $n = 2934$ respondents; Willing: $n = 1714$ respondents; Hesitant: $n = 280$ respondents) Tanzania (**d**, National estimate: $n = 2196$ respondents; Willing: $n = 1009$ respondents; Hesitant: $n = 407$ respondents), Uganda (**e**, National estimate: $n = 1872$ respondents; Willing: $n = 1677$ respondents; Hesitant: $n = 134$ respondents). The bottom and top border of each box plot demarcate the 25th and 75th percentile, respectively, while the line inside the box marks the median. Whiskers extend from the bottom (top) end of the box to the lowest (highest) value within 1.5 times of the inter-quartile range. Dots outside the whiskers show outlier values. The red dotted line is the estimated mean for those hesitant to get vaccinated and the blue line the estimated mean for those willing to get vaccinated. The yellow line is the estimated national level acceptance for COVID-19 vaccines.

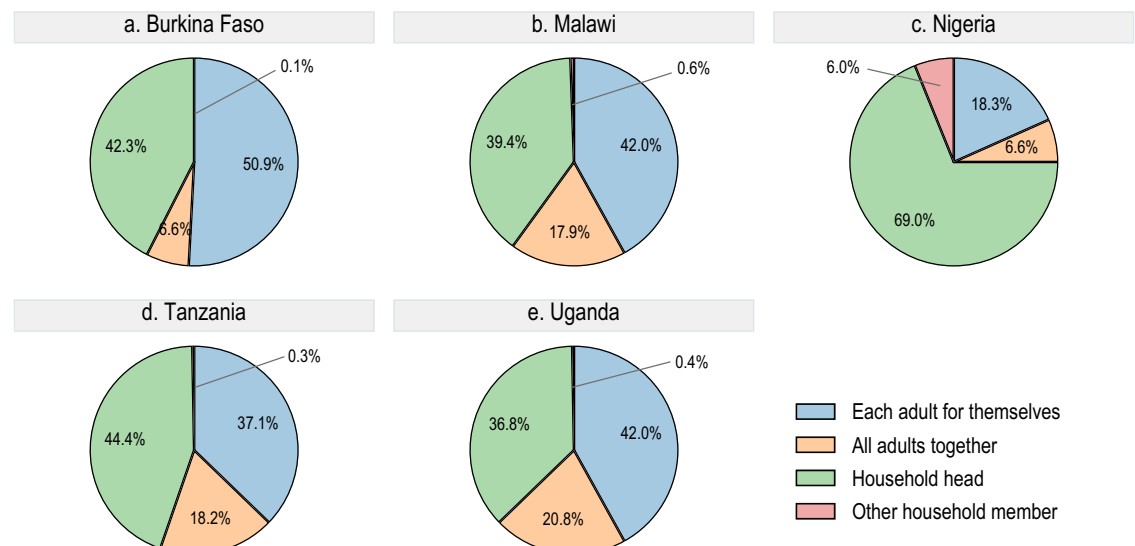

**Fig. 6 Decision maker about vaccine uptake among adult household members.** Burkina Faso (**a**, $n = 1744$ respondents), Malawi (**b**, $n = 1343$ respondents), Nigeria (**c**, $n = 2386$ respondents), Tanzania (**d**, $n = 2131$ respondents), Uganda (**e**, $n = 1870$ respondents). Blue area = each adult decides for themselves, orange area = all adults decide together, green area = household head decides, red area = other household member decides.

There is overwhelming willingness among those vaccinated to do so: In Malawi (80.7%, CI 75.0–86.4%), Tanzania (76.4%, CI 70.7–82.2%), and Nigeria (72.1%, CI 67.3–76.9%), more than 7 out of 10 vaccinated individuals would be "very likely" to encourage others to get vaccinated (6 in 10 in Uganda: 59.3%, CI 55.8 to 62.9). In Burkina Faso, most are either very likely (22.9%, CI 18.2–27.7%) or somewhat likely (58.4%, CI 52.9–63.8%).

## Discussion

Since the start of efforts to vaccinate the world population for COVID-19, vaccine access has been highly unequal across the globe[6,37,38]. This has given rise to large regional disparities in COVID-19 vaccine coverage that persist until today. Sub-Saharan Africa in particular trails the rest of the world which threatens to exacerbate health and economic inequities within the region and on a global level. Increasing COVID-19 vaccine coverage in Sub-Saharan Africa is also called for from an epidemiological viewpoint. To prevent future mutations of the virus and contain COVID-19 globally, broad coverage among the 1.17 billion people living in Sub-Saharan Africa is vital.

In this study, we use data from five national phone surveys conducted by each country's national statistical agency to provide comprehensive, cross-country comparable, and timely insights into the factors that hold back the progress of vaccination campaigns. Our findings on vaccine hesitancy, uptake, local barriers of access, and possible promoters of vaccine demand come at a critical moment for vaccination campaigns: Vaccination campaigns are now underway across the region but have generally failed to reach beyond a fraction of the population.

Our findings update and expand on results from previous cross-country studies many of which were conducted before the wider availability of vaccines[18–22,39] and add an up-to-date empirical basis to recent discussions of vaccine hesitancy and last-mile delivery barriers in the uptake of vaccines[7–12,14,16,17,22,40–42]. Our findings particularly complement recent longitudinal evidence that finds sustainedly high levels of vaccine acceptance in Sub-Saharan Africa[42]. At the same time, this literature caveats that individual attitudes appear to change frequently over time and that high levels of acceptance are a necessary but not sufficient condition for high vaccine take-up[7,42]. For vaccine coverage rates to catch up to acceptance rates for COVID-19 vaccines in Sub-Saharan Africa, recent studies have emphasized the importance of ongoing outreach efforts as well as investments in last-mile delivery capacity[6,31–35]. Our paper provides cross-country evidence that can inform the messaging and mode of such outreach efforts and identifies factors that currently hinder widespread access to vaccines at the local level.

Our analysis addresses a broad range of issues and information gaps and spans all five dimensions of vaccine hesitancy that have been hypothesized to underlie the low take-up of vaccines in Sub-Saharan Africa and elsewhere:[23] Trust in the safety and efficacy of vaccines (confidence), perceived severity of COVID-19 and risk to fall ill (complacency), access to COVID-19 vaccines (convenience), the level and sources of information (communications), and sociodemographic characteristics (context, which is explored in greater depth in another study[42]). Furthermore, we provide insights into the last-mile delivery barriers that have kept vaccine-willing people from accessing vaccines and opportunities to reach out to those who are hesitant. As such, our findings can inform strategies that national vaccination campaigns may pursue to turn vaccines into vaccinations in Africa.

Our study has several limitations related in particular to phone survey data collection on vaccination. Our phone survey samples suffer from varying degrees of sample selection at the household-level due to under-coverage, non-response, and attrition, and

within the household arising from the purposive selection of respondents[29,30]. We use recalibrated sampling weights to reduce the effects of sample selection, but our estimates cannot be considered nationally representative and likely over-represent better-off and urban households, and older, better educated, male individuals (Supplementary Table 16). Furthermore, survey data, regardless of mode (phone, face-to-face, online), necessarily relies on respondent self-reporting which is susceptible to respondents' incentives, misreporting, and misperceptions. Lastly, while our estimates cover almost a third of the population of Sub-Saharan Africa and a geographically, culturally, economically, and socially diverse set of countries, they need not be representative of all countries in the region. Relatedly, our study covers a broad domain of policy options that emerge across the countries we study, but their specific implementation should be informed by dedicated (case) studies that can adequately account for the local context and complexity of the issue at hand.

We find that in our study countries a majority remains willing to get vaccinated but that hesitancy among those unvaccinated is a non-negligible issue. As vaccine coverage in much of Sub-Saharan Africa is still below 30 percent, vaccination campaigns should focus first on getting those who are willing but yet unvaccinated to take up the vaccine. The main barriers keeping this group away from the vaccination sites are country-specific but commonly relate to the ease with which vaccines can be accessed within communities. Therefore, it is indispensable that vaccination sites become more widespread at the local level[41]. High opportunity costs to accessing vaccines and local supply constraints also resonate with recent calls for more vaccine equity and an expansion of domestic vaccine manufacturing capacity in the region[6,7,17,37,38,41,43–46]. The benefits of such investments would extend beyond the COVID-19 pandemic[44].

Furthermore, a continuation of communication campaigns about the ongoing risk of COVID-19 and safety of vaccines will be pivotal: We find that the protection vaccines afford to one's own health is the main reason why people take up the vaccine and that hesitancy mostly relates to concerns about the vaccine's side effects[36]. As this was already the main concern among the hesitant in 2020[18–20,42], national vaccination campaigns should double-down on their efforts to emphasize that the relative benefits of COVID-19 vaccination outweigh the associated risks in almost all cases[47]. As longitudinal studies of COVID-19 vaccine hesitancy in Sub-Saharan Africa have pointed to the malleability of attitudes[42], closing existing information gaps may help change the stances of those who lack a complete picture of the relative benefits of vaccination.Concerning the means through which the population acquires information on COVID-19 vaccination, we find that mass media devices are widely used by respondents. Their widespread use is further corroborated by estimates from the pre-COVID-19 nationally representative household surveys that serve as the sampling frames for the HFPS: The data suggest that between 70% (Malawi) and 94% (Burkina Faso) of households own a radio or mobile phone (85% in Nigeria, 87% in Uganda, estimates not available in Kenya and Tanzania). Radio broadcasts are therefore a promising avenue through which mass communication campaigns could reach their target population. For example, such campaigns could follow the example of an initiative by Farm Radio International, a network of radio stations across 41 Sub-Saharan African countries, that ran radio programs in select countries to better inform the population and dispel myths about COVID-19 during the pandemic[48]. Similarly, Liberian radio stations broadcast health information and interviews with survivors during the Ebola pandemic[49]. An important advantage of these initiatives is that they leverage an established and trusted medium and can

take into account the local context (for example, by conveying geographically targeted information in a local language). Our results also suggest that medical professionals are highly trusted among both the willing and hesitant. Vaccination campaigns, including outreach efforts over radio, could employ endorsements by this group in order to sway the opinions of the hesitant.

Lastly, we find that the social context informs vaccine attitudes, as perceptions about the general acceptance of COVID-19 and within-household power dynamics matter. Changing public perceptions about the true support for vaccines in the country and creating a positive social norm around vaccination could be effective measures. Those already vaccinated could be mobilized as ambassadors that can promote vaccine demand at the community level and improve grassroots monitoring of last-mile delivery barriers. Pilot initiatives to this effect are underway for example in Côte d'Ivoire, Malawi, Senegal, South Africa, and Zimbabwe[50,51]. As part of these initiatives, volunteers are trained as "health ambassadors" and tasked with spreading reliable and accurate information about the importance of vaccination and asked to report on local bottlenecks in vaccine delivery. Furthermore, convincing key members of the household, in particular the household head, of the need for vaccination may positively affect the take-up of vaccines by the remaining household members. These findings are in line with recent experimental evidence from Zambia[36].

Vaccine hesitancy, as defined by the WHO in a recent position paper, describes "a motivational state of being conflicted about, or opposed to, getting vaccinated"[52]. This definition acknowledges that actual uptake of vaccines is distinct from stated intentions and is affected by many more factors. The strategies we recommend – reducing the opportunity costs of getting vaccinated, encouraging vaccination by emphasizing the health benefits of vaccines, and leveraging trusted ambassadors – are thus worthwhile investments to increase take-up even among those that are currently hesitant. This is because these policies can reduce structural, informational, and social barriers to vaccination that influence take-up in addition to each individual's hesitancy status.

With COVID-19 vaccines becoming more widely available but a large majority of African countries far off the WHO's target to fully vaccinate 70 percent of the population by June 2022, now is the time to push for increased vaccine take-up in Sub-Saharan Africa. With a renewed focus on creating a positive social norm around COVID-19 vaccination, messaging that leverages trusted and accessible information sources and channels, and more easily accessible vaccination sites at the community level, countries can speed up this process and successfully turn vaccines into vaccinations.

## Data availability

The source data required to replicate the tables and figures used in this study as well as the full questionnaires have been made publicly available on the Harvard Dataverse[53]. The raw data is available through the World Bank's microdata library:[24] • Burkina Faso (https://microdata.worldbank.org/index.php/catalog/3768). • Kenya (https://microdata.worldbank.org/index.php/catalog/3774 - data available upon request with country team). • Malawi (https://microdata.worldbank.org/index.php/catalog/3766). • Nigeria (https://microdata.worldbank.org/index.php/catalog/4444). • Tanzania (https://microdata.worldbank.org/index.php/catalog/4542). • Uganda (https://microdata.worldbank.org/index.php/catalog/3765). • The data for Fig. 1 is also available from the Our World in Data GitHub page[31]. • Any remaining data are available from the corresponding author on reasonable request.

## Code availability

Authors used Stata/MP 17.0 for data analysis. The code to replicate all figures and tables is publicly available in the Harvard Dataverse[53].

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

## Acknowledgements

This paper received funding support from the World Bank Research Support Budget grant "Understanding and estimating COVID-19 vaccination attitudes, uptake, and barriers in Sub-Saharan Africa" and the Global Financing Facility. The funders had no role in the study design, implementation, or analysis. We are grateful for support in designing the survey instrument and collecting the data by the country teams: Marco Tiberti for Burkina Faso; Wilbert Drazi Vundru for Malawi; Akiko Sagesaka, Amparo Palacios-Lopez, Ivette Contreras Gonzalez, and Gbemisola Oseni for Nigeria; and Akuffo Amankwah for Tanzania; Giulia Ponzini and Frederic Cochinard for Uganda; Antonia Delius and Caleb Leseine Gitau for Kenya. We are furthermore grateful for comments and support by Gero Carletto, Talip Kilic, and Kevin McGee and comments and feedback in the development of the survey instruments by the members of the World Bank Development Data Group's HFPS Questionnaire Working Group.

## Author contributions

Conceptualization: P.W., Y.M., A.Z; Data curation: Y.M., S.K; Formal Analysis: Y.M.; Funding acquisition: P.W., Y.M., A.Z; Methodology: P.W., Y.M., S.K., A.Z; Project administration: A.Z., S.K.; Software: Y.M.; Supervision: P.W., A.Z.; Validation: P.W.; Visualization: Y.M.; Writing – original draft: P.W., Y.M.; Writing – review & editing: P.W., Y.M., A.Z., S.K.

## Competing interests

The authors declare no competing interests.
