## [Peer Review File · Communications Medicine]

Reviewers' comments:

Reviewer #1 (Remarks to the Author):

Overall the study adds value to the continued discussion around vaccine hesitance versus access. As the number of publications focused on access are few this is very important. Results are given the largest amount of space but there is less detail in terms of practical recommendations or policy recommendations which was anticipated from the Abstract and the data could lead to some good operational suggestions. Does not clearly link to other evidence in terms of access to vaccinations / health care which could have strengthened the article.

Is a useful manuscript for public health practitioners especially given the scale of the data, also gave good recommendations for future study/ research.

38 – figure is not clear due to number of data points. Suggest splitting into 2 charts 1 to show the 4 income brackets and 1 to show the countries Vs global uptake.

45 – This is not entirely new evidence as there are advocacy efforts ongoing to change the global narrative and academic researchers who have also been adding to the pool of knowledge. Suggest to re frame this.

54 – Suggest to give the actual time frame November 2021 to August 2022

55- Suggest change of wording use of “supersedes” would imply this study replaces other data rather than adding to the community of knowledge.

75 – Need to give a break-down of respondents by sex, this was not contained in the supplementary information and means this suggestion cannot be evidenced.

84 – Would be interesting to discuss more the common themes in terms of unwillingness. Linked to 136-148. Without knowing base line knowledge of the respondents this could still be an issue of access to information rather than real hesitance. This has implications for policy and implementation by health ministries in terms of communication campaigns.

152 – 161 - Could be useful to mention coverage of social media / mobile phone ownership and radio coverage to add value to the results.

167 – Suggest to change wording from “people that could encourage them to get vaccinated” to “people that could be vaccine ambassadors” for consistency.

178 – Would add value if radio coverage survey, mobile phone ownership statistics etc could be referenced. Suggest using Demographic health surveys, Ministries of information reports etc. your estimate here is based only on respondents who represent only members of the population in areas of phone coverage / ownership.

191-192 – suggest re wording / clarification. The question here does not address community beliefs about vaccinations, it addresses respondent perception of vaccine acceptance.

224-228 – Suggest to consider reframing, this paragraph could be read as giving a negative assessment of SSA vaccination efforts without mentioning issues of availability of vaccines to countries.

233 – Suggest clarification regarding vaccination availability – this may be true at a central level (in each country) but does not consider the other aspects of vaccine availability. Suggest removing the term widely available which disregards internal movement of vaccine doses and commodities.

242 – Consider specifying the socio-demographic context as the break-down of respondents per context, sex, economic group, education level etc were not utilised.

246-255 - This is a good clear identification of limitations, useful for the reader.

269 – Suggest evidencing this. Or clarifying “among respondents”

290 – Suggest to clarify whether the questions were already part of the HFPS or were added for the purpose of this study. Slightly unclear.

312 – Suggest to add how the survey was introduced including were general demographic information captured?

321 – 331 – this is clear and concise and gives a good reflection of why the questions were selected.

Reviewer #2 (Remarks to the Author):

Thanks a lot for the invitation to review this timely and important manuscript.

In the current study, Philip Wollburg et al. conducted a large phone-based survey study on COVID-19 vaccine acceptance and its determinants in six sub-Saharan African countries. The importance of this study is related to previous evidence of relatively high rates of COVID-19 vaccine hesitancy in several African countries particularly in West Africa, and the relative lack of studies addressing this objective in the region using a systematic and robust approach. Additionally, the current study findings can help in tailoring the public health intervention measures to better suit the region considering that COVID-19 vaccine hesitancy is place and culture specific phenomenon.

The major results pointed to variable but noticeable prevalence of vaccination hesitancy per country with inconvenience as a major determinant of lower vaccine acceptance. Furthermore, the trust in health professionals and radio broadcasts as indicated in the study should be taken into account in the efforts aiming to promote vaccine uptake in the countries of the region.

Overall, the manuscript is well written, with robust methodology and the results were presented clearly.

I have a few minor comments that hopefully can help the authors to improve the final manuscript as follows:

1. In the Introduction and the Discussion sections, the authors can benefit from a recently published opinion article regarding the COVID-19 vaccine landscape in Africa:

<https://doi.org/10.3389/fimmu.2022.955168>

2. In Figure 1, which was highly valuable, the authors can benefit from improving the clarity of different lines to help the readers in tracking different lines.

3. In Figure 3, one data point is missing for the item “Facility not disability friendly” for Malawi. Please double-check this issue.

Thank you!

Reviewer #3 (Remarks to the Author):

The article by Wollburg and colleagues explored how to turn vaccines into vaccination in sub-Saharan Africa by comparing data from countries in East and West Africa.

Vaccine acceptance or willingness to receive the vaccine was very high and they identified access to vaccination as the main driver for vaccine uptake among those willing to be vaccinated.

Overall, this was a great study with the main strength being simultaneous comparison of data across different countries in two regions.

Below are some comments

1. Even though the vaccine hesitancy was relatively lower in the current study compared with previously reported data from various countries, the authors could discuss their results in context considering that vaccine hesitancy can vary across time and some of the studies were done before or at the time of vaccine introduction and perceptions may have changed given the high level of population immunity now from either vaccines or natural immunity from infection. The results of hesitant population could also be discussed taking into account the updated definition of vaccine hesitancy (WHO, BeSD, 2022)

2. In Table 1 and supplementary information, the sample size for RDD for Kenya is almost 93 Million. I looked through the text and most of the other information in other tables including the supplementary data and was not able to understand the justification of that figure and how the other data from Kenya align to it. The figure is almost more than 150% of the country's population. If the authors could somehow provide a line or two in the text to discuss that number it might be helpful.

3. In line 266 where the authors recommend doubling-down on the efforts to assure the population about safety of the vaccine, I am not sure whether that is necessary given the side effects are still a concern on the safety of the vaccines.

Reviewer #4 (Remarks to the Author):

Overview: This descriptive paper presents results from multiple SSA countries about vaccine hesitancy and the 5 attributes. It aims to inform policy by its results. While the results are novel, and of interest, the target population the results claim to cover are grossly over-stated. This over-statement could lead country governments to create policies that, in some cases, exclude the experience of up to 40% of their population (non-phone owners), who we know are very different from phone owners. This paper needs to be re-written to appropriately address the limitations of phone surveys.

Comments:

The limitations of phone surveys are not yet clear enough to the general public health research audience to not be explicit about said limitations in a paper. This paper is written without sufficient explanation of the biases that phone surveys present. Health outcomes are consistently over-estimated in a positive direction given phone owners are more educated, urban, male and likely to speak English/French than their non-phone owning counterparts. This bias is exacerbated by the study design that sampled "who was selected to be knowledgeable of the affairs of the household and its members to

309 provide reliable responses, though this selection overrepresents certain population groups" The article in PlosONE by a similar author group, for the same phone surveys presented in this article states "reweighting fails to overcome the biases in most cases as the difference in means remain statistically significant for most of the outcomes of interest." Therefore the claim that these results "are representative of a population of 415 million people" is grossly over-stated. The manuscript has to be up-front about these biases and cannot simply link readers to articles that explain this, because again, the average reader does not yet have enough knowledge of limitations of phone surveys in Africa.

The conclusion must also address this limitation very clearly.

The manuscript should also states response rates (using AAPOR definitions). Given the already lengthy supplementary file attachment, a table of response rates would be reasonable to add as would tables of demographic information about respondents compared to the FTF survey.

Given RDD sampling was used in Kenya and RDD presents more sampling error than following up with a FTF survey, the finding that vaccine acceptance is near-universal in Kenya is not surprising.

Again, the results that most people have knowledge of campaigns and that knowledge of campaigns is not preventing uptake reflects findings of those who are better connected, more educated. In countries where phone ownership is low (presumably Malawi, Tanzania, certainly Uganda at 60% of the population), these results are particularly skewed.

page 3, line 66: comparisons to other studies should be in discussion section, not directly in results section.

page 10, lines 246- 255: the discussion of limitations of phone surveys is insufficient. you already know who is excluded, and have published on it. further more, calling for more research on the implications of the existing biases is not an appropriate call to action. you need to directly state who is excluded/ who this actually represents. if governments use your results to create policy, they will be creating policy that does not address what the poorest poor, women, rural people need - those who are also at greatest risk of not being vaccinated.

"The implications of these issues for findings on vaccine hesitancy should be the subject of future research as should be the reliability of survey data on vaccination in the context of COVID-19.32–34"

Methods: need more about sampling. also lacking statistical analysis explanation. needed even if paper is descriptive only.

Reviewers' comments:

Reviewer #1 (Remarks to the Author):

Overall the study adds value to the continued discussion around vaccine hesitance versus access. As the number of publications focused on access are few this is very important. Results are given the largest amount of space but there is less detail in terms of practical recommendations or policy recommendations which was anticipated from the Abstract and the data could lead to some good operational suggestions. Does not clearly link to other evidence in terms of access to vaccinations / health care which could have strengthened the article.

Is a useful manuscript for public health practitioners especially given the scale of the data, also gave good recommendations for future study/ research.

Response: *Thank you for your thoughtful comments that we feel have significantly improved the manuscript. In response to your feedback, we are now linking more explicitly to other research on COVID-19 vaccination in Sub-Saharan Africa throughout the Discussion section. We have also bolstered the policy recommendations with more concrete examples for how radio broadcasts and vaccine ambassadors can be leveraged. Lastly, we now caveat that, while a cross-country study like ours can cover a broad range of high-level policy options, the issue of improving vaccine coverage in Sub-Saharan Africa is complex. The specific implementation of the different policies we outline should thus be informed by dedicated studies that can more fully account for the local context.*

We are responding to your remaining comments point-by-point below and hope to have addressed your remaining concerns to your satisfaction.

38 – figure is not clear due to number of data points. Suggest splitting into 2 charts 1 to show the 4 income brackets and 1 to show the countries Vs global uptake.

Response: *In line with the suggestions by Reviewer #2, we have attempted to make the lines in the graph more distinct so that regional and country-level figures are more easily distinguishable.*

45 – This is not entirely new evidence as there are advocacy efforts ongoing to change the global narrative and academic researchers who have also been adding to the pool of knowledge. Suggest to re frame this.

Response: *We fully agree that our study is part of a larger effort to address the COVID-19 vaccination gap in Sub-Saharan Africa, including studies published by other academic researchers. We have adjusted the phrasing to reflect the fact that we contribute to this growing body of evidence and literature.*

54 – Suggest to give the actual time frame November 2021 to August 2022

Response: *This has been adjusted.*

55- Suggest change of wording use of “supersedes” would imply this study replaces other data rather than adding to the community of knowledge.

Response: *We have adjusted the phrasing here and in line 235.*

75 – Need to give a break-down of respondents by sex, this was not contained in the supplementary information and means this suggestion cannot be evidenced.

Response: *We have included a breakdown of the number of respondents by sex in Table A2.*

84 – Would be interesting to discuss more the common themes in terms of unwillingness. Linked to 136-148. Without knowing base line knowledge of the respondents this could still be an issue of access to information rather than real hesitancy. This has implications for policy and implementation by health ministries in terms of communication campaigns.

Response: *We fully agree that inadequate information can contribute to vaccine hesitancy. We also agree that we cannot ascertain whether hesitancy as observed in our study reflects information gaps or genuine refusal of the vaccine in the face of full information. In our study, we do not further distinguish between those who answer “no” and those that answer “not sure” when asked whether they are planning to get vaccinated. One reason is that the share of those answering “not sure” is small (between 18% and 2% of those not yet vaccinated) rendering our sample size insufficient to reliably disaggregate the analysis in lines 136 – 148 between the two groups. Another reason is that we feel like even answering “no” may not reflect outright vaccine refusal and the distinction between both answer options is not necessarily clear cut – this seems to be supported by the fact that the majority of our respondents (including many of those answering “no”) say they could be convinced to change their stance by a trusted vaccine ambassador. In line with the journal’s guidelines, we have incorporated your argument in the Discussion section.*

152 – 161 - Could be useful to mention coverage of social media / mobile phone ownership and radio coverage to add value to the results.

Response: *Thank you for this suggestion. We have provided a reference for the wide dissemination of radio as an information medium in Sub-Saharan Africa. In line with your comment below, we are also now including ownership figures for radios and mobile phones based on nationally representative household surveys in the discussion section.*

167 – Suggest to change wording from “people that could encourage them to get vaccinated” to “people that could be vaccine ambassadors” for consistency.

Response: *We have adjusted the phrasing for consistency.*

178 – Would add value if radio coverage survey, mobile phone ownership statistics etc could be referenced. Suggest using Demographic health surveys, Ministries of information reports etc. your estimate here is based only on respondents who represent only members of the population in areas of phone coverage / ownership.

Response: *Thank you for this suggestion. We are now reporting coverage figures for mass media devices (radios and/or mobile phones) based on the nationally representative, pre-COVID face-to-face household*

surveys that serve as the sampling frames for our phone surveys (being based on face-to-face surveys, these estimates come from the general population, not just those who own phones). These figures confirm their widespread reach in the general population of our study countries. We have taken the liberty to include these figures in the discussion section when discussing the role and reach of radio broadcasts.

191-192 – suggest re wording / clarification. The question here does not address community beliefs about vaccinations, it addresses respondent perception of vaccine acceptance.

Response: *We agree and have corrected the phrasing.*

224-228 – Suggest to consider reframing, this paragraph could be read as giving a negative assessment of SSA vaccination efforts without mentioning issues of availability of vaccines to countries.

Response: *Thank you for your suggestion. We have re-written the paragraph in question to make it clear that it does not aim to criticize a lack of effort on the side of vaccination campaigns in SSA and that, conversely, those efforts have been hampered by the unequal distribution of vaccines across the globe.*

233 – Suggest clarification regarding vaccination availability – this may be true at a central level (in each country) but does not consider the other aspects of vaccine availability. Suggest removing the term widely available which disregards internal movement of vaccine doses and commodities.

Response: *As suggested, we have removed the term ‘widely available’ and adjusted the phrasing to reflect the fact that vaccination campaigns are now underway across the region without implying anything about the local availability of vaccines.*

242 – Consider specifying the socio-demographic context as the break-down of respondents per context, sex, economic group, education level etc were not utilised.

Response: *Thank you for your comment. To cover the context theme as defined by Razai et al. (2021), we include a breakdown of vaccine acceptance and the barriers of access by gender and urban/rural residence and discuss the role of local communities and trusted ambassadors in encouraging vaccine take-up. We agree that a comprehensive analysis of the socio-demographic context is necessarily much broader and are now referring readers to a companion paper in which we analyze the socio-demographic context of vaccine hesitancy in much greater depth.*

246-255 - This is a good clear identification of limitations, useful for the reader.

Response: *Thank you for your assessment.*

269 – Suggest evidencing this. Or clarifying “among respondents”

Response: *We have adjusted the phrasing as suggested.*

290 – Suggest to clarify whether the questions were already part of the HFPS or were added for the purpose of this study. Slightly unclear.

Response: *We have clarified that the survey module on vaccination that we draw on was included specifically for this study. We have taken the liberty to point this out under the “Survey instrument and variables” instead of the “Data” sub-heading.*

312 – Suggest to add how the survey was introduced including were general demographic information captured?

Response: *We have clarified that the HFPS include a roster with demographic information on all household members in each survey round under the “Data” sub-heading. We have also included information on the introduction text that enumerators read out to respondents before administering the COVID-19 vaccination module as part of the larger HFPS interview under the “Survey instrument and variables” sub-heading.*

321 – 331 – this is clear and concise and gives a good reflection of why the questions were selected.

Response: *Thank you for your assessment.*

Reviewer #2 (Remarks to the Author):

Thanks a lot for the invitation to review this timely and important manuscript.

In the current study, Philip Wollburg et al. conducted a large phone-based survey study on COVID-19 vaccine acceptance and its determinants in six sub-Saharan African countries. The importance of this study is related to previous evidence of relatively high rates of COVID-19 vaccine hesitancy in several African countries particularly in West Africa, and the relative lack of studies addressing this objective in the region using a systematic and robust approach. Additionally, the current study findings can help in tailoring the public health intervention measures to better suit the region considering that COVID-19 vaccine hesitancy is place and culture specific phenomenon.

The major results pointed to variable but noticeable prevalence of vaccination hesitancy per country with inconvenience as a major determinant of lower vaccine acceptance. Furthermore, the trust in health professionals and radio broadcasts as indicated in the study should be taken into account in the efforts aiming to promote vaccine uptake in the countries of the region.

Overall, the manuscript is well written, with robust methodology and the results were presented clearly.

Response: *Thank you for taking the time to provide us with comments on our manuscript and the kind feedback on our study. We have taken care to address your comments and reply point-by-point below.*

I have a few minor comments that hopefully can help the authors to improve the final manuscript as follows:

1. In the Introduction and the Discussion sections, the authors can benefit from a recently published

opinion article regarding the COVID-19 vaccine landscape in Africa:

<https://doi.org/10.3389/fimmu.2022.955168>

Response: Thank you for pointing us to this important resource. The aspect of improved vaccine equity and enhanced local manufacturing capacity had previously not featured in our manuscript and we now pick up this point in the Discussion section along with other relevant literature.

2. In Figure 1, which was highly valuable, the authors can benefit from improving the clarity of different lines to help the readers in tracking different lines.

Response: Thank you making us aware of this. Your point is in line with a comment by Reviewer #1 and we have improved the visibility of the different lines in Figure 1 in response.

3. In Figure 3, one data point is missing for the item “Facility not disability friendly” for Malawi. Please double-check this issue.

Response: Thank you for catching this. We have included the missing data point which sits at 0%.

Reviewer #3 (Remarks to the Author):

The article by Wollburg and colleagues explored how to turn vaccines into vaccination in sub-Saharan Africa by comparing data from countries in East and West Africa.

Vaccine acceptance or willingness to receive the vaccine was very high and they identified access to vaccination as the main driver for vaccine uptake among those willing to be vaccinated.

Overall, this was a great study with the main strength being simultaneous comparison of data across different countries in two regions.

Response: Thank you for your kind feedback and for your comments that have helped improve the manuscript further. We hope to have addressed your comments to your satisfaction and reply point-by-point below.

Below are some comments

1. Even though the vaccine hesitancy was relatively lower in the current study compared with previously reported data from various countries, the authors could discuss their results in context considering that vaccine hesitancy can vary across time and some of the studies were done before or at the time of vaccine introduction and perceptions may have changed given the high level of population immunity now from either vaccines or natural immunity from infection. The results of hesitant population could also be discussed taking into account the updated definition of vaccine hesitancy (WHO, BeSD, 2022)

Response: Thank you for pointing this out. We now discuss the issue of potential changes in vaccine attitudes over time and link to a companion paper that studies longitudinal dynamics in vaccine hesitancy across much of the same sample used here. We also acknowledge more explicitly that vaccine acceptance is a necessary but not sufficient condition for vaccine take-up and that opportunity costs as well as the perceived urgency of vaccination play a role in this regard. Lastly, we thank you for having pointed us to the updated definition of vaccine hesitancy by the WHO. We have incorporated this into our discussion.

2. In Table 1 and supplementary information, the sample size for RDD for Kenya is almost 93 Million. I looked through the text and most of the other information in other tables including the supplementary data and was not able to understand the justification of that figure and how the other data from Kenya align to it. The figure is almost more than 150% of the country's population. If the authors could somehow provide a line or two in the text to discuss that number it might be helpful.

Response: We agree that our previous discussion of the sampling strategy lacked some detail and are now providing a more extensive description in the methods section. As for the RDD sample in Kenya, the figure in question (93 million) is the universe of all phone numbers from which a random subsample was drawn and subsequently called for the Kenya survey. This number is larger than the population of Kenya because the phone numbers come from three mobile operators, they include phone numbers that are currently unassigned, and it is possible that one individual has more than one number. We provided additional information on the Kenya RDD sample in the Sampling section.

3. In line 266 where the authors recommend doubling-down on the efforts to assure the population about safety of the vaccine, I am not sure whether that is necessary given the side effects are still a concern on the safety of the vaccines.

Response: We of course agree that COVID-19 vaccines are not free of any risks or potential side effects. We have thus adjusted the phrasing of the paragraph in question and now call for an emphasis of the relative benefits of vaccination that outweigh the associated risks for the vast majority of the population.

Reviewer #4 (Remarks to the Author):

Overview: This descriptive paper presents results from multiple SSA countries about vaccine hesitancy and the 5 attributes. It aims to inform policy by its results. While the results are novel, and of interest, the target population the results claim to cover are grossly over-stated. This over-statement could lead country governments to create policies that, in some cases, exclude the experience of up to 40% of their population (non-phone owners), who we know are v different from phone owners. This paper needs to be re-written to appropriately address the limitations of phone surveys.

Response: Thank you for taking the time to review our study and for your thoughtful comments on sampling and survey methodology. We have incorporated your comments in the revised manuscript and paid close attention in particular to statements regarding the representativeness of the survey data.

Comments:

The limitations of phone surveys are not yet clear enough to the general public health research audience to not be explicit about said limitations in a paper. This paper is written without sufficient explanation of

the biases that phone surveys present. Health outcomes are consistently over-estimated in a positive direction given phone owners are more educated, urban, male and likely to speak English/French than their non-phone owning counterparts. This bias is exacerbated by the study design that sampled "who was selected to be knowledgeable of the affairs of the household and its members to 309 provide reliable responses, though this selection overrepresents certain population groups" The article in PlosONE by a similar author group, for the same phone surveys presented in this article states "reweighing fails to overcome the biases in most cases as the difference in means remain statistically significant for most of the outcomes of interest." Therefore the claim that these results "are representative of a population of 415 million people" is grossly over-stated. The manuscript has to be up-front about these biases and cannot simply link readers to articles that explain this, because again, the average reader does not yet have enough knowledge of limitations of phone surveys in Africa. The conclusion must also address this limitation very clearly.

Response: *Thank you for this comment whose main point is well taken. We have taken several steps to address this comment. First, we removed the sentence "are representative of a population of 415 million people" and retain only a reference to the population size of the study (as we contend this is relevant information on the country coverage of our study) without claiming representativeness of the survey samples. Second, we now discuss the limitations of the study's representativeness in the Discussion section. Here we state that the estimates cannot be considered nationally representative and also discuss the steps we take to improve the representativeness of our estimates relative to unweighted estimates. Third, we significantly expanded the Method section on Sampling and sample representativeness in which we discuss these same issues in greater detail.*

The manuscript should also states response rates (using AAPOR definitions). Given the already lengthy supplementary file attachment, a table of response rates would be reasonable to add as would tables of demographic information about respondents compared to the FTF survey.

Response: *We added responses rates in Table 1: Sample Size and discuss them also in the methods section. We added a table on demographic information about respondents compared to FTF population (where applicable and available to us) in Table A17.*

Given RDD sampling was used in Kenya and RDD presents more sampling error that following up with a FTF survey, the finding that vaccine acceptance is near-universal in Kenya is not surprising.

Response: *RDD does indeed present more sampling error, so that the RDD-based estimates are less representative, a fact that we now state in the Methods section.*

Again, the results that most people have knowledge of campaigns and that knowledge of campaigns is not preventing uptake reflects findings of those who are better connected, more educated. In countries where phone ownership is low (presumably Malawi, Tanzania, certainly Uganda at 60% of the population), these results are particularly skewed.

Response: *We have included coverage rates in Table 1 in the Methods section to give readers a sense of the share of households that were left out in each country because they did not have access to a phone. We also expanded our discussion of under-coverage in the Methods section.*

page 3, line 66: comparisons to other studies should be in discussion section, not directly in results section.

Response: *This is noted and rectified.*

page 10, lines 246- 255: the discussion of limitations of phone surveys is insufficient. you already know who is excluded, and have published on it. further more, calling for more research on the implications of the existing biases is not an appropriate call to action. you need to directly state who is excluded/ who this actually represents. if governments use your results to create policy, they will be creating policy that does not address what the poorest poor, women, rural people need - those who are also at greatest risk of not being vaccinated.

"The implications of these issues for findings on vaccine hesitancy should be the subject of future research as should be the reliability of survey data on vaccination in the context of COVID-19.32–34"

Response: *We expanded the Discussion on the limitations of the study's representativeness, and included information on which households and individuals are over-represented. The discussion also links to Table A17 which shows sociodemographic characteristics of respondents relative to the adult population.*

Methods: need more about sampling. also lacking statistical analysis explanation. needed even if paper is descriptive only.

Response: *We have significantly expanded the section on Sampling and representativeness following your recommendations. We added a short section on statistical analysis (Estimation).*

REVIEWERS' COMMENTS:

Reviewer #1 (Remarks to the Author):

Thank you for this re submission and for taking note of the reviewer comments. the changes made on the whole make the manuscript more robust.

The research certainly adds to the community of knowledge in respect to COVID-19 vaccinations and has other information which could have cross cutting use for health campaigns generally.

thank you

Reviewer #2 (Remarks to the Author):

Thanks for properly addressing all the previous comments.

Wishing you the best!

Reviewer #3 (Remarks to the Author):

My comments were addressed

Reviewer #4 (Remarks to the Author):

Appreciate the updates.